# Transcriptomic Analysis Reveals the Role of Long Non-Coding RNAs in Response to Drought Stress in Tibetan Hulless Barley

**DOI:** 10.3390/biology14070737

**Published:** 2025-06-20

**Authors:** Zitao Wang, Yue Fang, Qinyue Min, Kaifeng Zheng, Yanrong Pang, Jinyuan Chen, Feng Qiao, Shengcheng Han

**Affiliations:** 1Key Laboratory of Biodiversity Formation Mechanism and Comprehensive Utilization of the Qinghai-Tibet Plateau in Qinghai Province, School of Life Sciences, Qinghai Normal University, Xining 810008, China; 18587712819@163.com (Z.W.); 18085839170@163.com (Y.F.); m1nq1nyue@163.com (Q.M.); 20211027@qhnu.edu.cn (J.C.); 2College of Life Sciences, Beijing Normal University, Beijing 100875, China; kaifeng_zheng@mail.bnu.edu.cn (K.Z.); 202321200016@mail.bnu.edu.cn (Y.P.); 3Academy of Plateau Science and Sustainability of the People’s Government of Qinghai Province & Beijing Normal University, Qinghai Normal University, Xining 810016, China

**Keywords:** long noncoding RNAs, protein-coding genes, transcriptome, drought tolerance, Tibetan hulless barley

## Abstract

The high-altitude ecosystems of the Qinghai-Xizang (Tibet) Plateau (QTP) endowed Tibetan hulless barley (*Hordeum vulgare* var. *nudum*) with remarkable resistance to abiotic stresses, making it an excellent crop model for exploring the drought-tolerance mechanism. Here, we characterized all long noncoding RNAs (lncRNAs) with the RNA sequencing (RNA-seq) data from drought-tolerant (Z772) and drought-sensitive (Z013) cultivars subjected to varying drought durations (0, 1, and 5 h). Then, we conducted serval analysis including structural features, differential expression, Weighted Gene Co-expression Network Analysis (WGCNA), Gene Ontology (GO) enrichment, and quantitative real-time polymerase chain reaction (qRT-PCR) analysis to identify the lncRNAs that may be involved in the drought stress response. We found that most lncRNAs were short, single-exon, and intergenic, and their nearby protein-coding genes (PCGs) were enriched in stress-response pathways. These results suggested that the specifically expressed lncRNAs appear to be directly related to the drought tolerance of the cultivars, and lncRNAs may regulate drought stress responses through both *cis*-regulatory and *trans*-regulatory mechanisms by controlling transcription factors (TFs) in Tibetan hulless barley.

## 1. Introduction

RNA can be categorized into two main types: protein-coding messenger RNAs (mRNAs) and noncoding RNAs (ncRNAs) [1]. Among the ncRNAs, long noncoding RNAs (lncRNAs) are a vital subclass involved in gene expression regulation and play critical roles in regulating the growth and development of all organisms [2,3]. LncRNAs are characterized by their extended length (≥200 nt) and classified into four major groups: intergenic, antisense, intronic, and potentially novel isoforms, based on their genomic position relative to mRNA [4,5]. Compared to mammalian lncRNAs, research on plant lncRNAs is still in its infancy, and their functional mechanisms require further investigation [6,7,8]. The lncRNA in rice, termed *VIVIpary*, has the capability to enhance seed dormancy [9]. In *Arabidopsis thaliana*, the *DROUGHT INDUCED lncRNA* (*DRIR*) functions as a positive regulatory factor, enhancing the tolerance to abiotic stresses [10]. During vernalization, *COOLAIR* and *COLDAIR* lncRNAs regulate the expression of *FLOWERING LOCUS C* (*FLC*), regulating floral timing [11]. To date, multiple studies have demonstrated that lncRNAs play diverse roles and promote gene expression through various mechanisms in response to external stimuli [12,13,14]. Although lncRNAs are known to regulate drought responses in model plants (*Arabidopsis thaliana*), their roles in crops for high-altitude agriculture remain poorly characterized. Elucidating these mechanisms could provide insights into crop adaptation to water-limited environments.

Of all abiotic stresses, drought stress demonstrates the highest prevalence, surpassing both temperature and salinity stress [15,16,17]. Furthermore, drought conditions have reduced food crop yields in arid regions, with impacts surpassing the cumulative effects of all plant diseases [18]. To combat drought stress, plants have developed complex molecular regulatory networks that enable a timely environmental response and adaptation [19,20,21]. Multiple studies have demonstrated that the drought response mechanisms in plants are linked to the abscisic acid (ABA) signaling pathways and lncRNAs [22,23]. High-throughput transcriptomic profiling under drought stress has systematically elucidated drought-responsive regulatory networks across diverse plant species [24,25]. LncRNA *MSTRG.6838* in maize shows co-downregulation with its target H^+^-ATPase subunit gene during drought stress, suggesting functional coregulation in stress response [26]. Li et al. [27] showed that rice develops drought stress memory through synergistic regulation between lncRNAs and abscisic acid (ABA), enhancing drought tolerance. It is clear that the growing demand for food has led to the search for more scientific ways to increase plant drought tolerance and increase crop yields [28]. Our research endeavors to unravel the intricate mechanisms through which lncRNAs confer drought tolerance, with the ultimate goal of elevating crop yields and enhancing grain quality.

Tibetan hulless barley, colloquially termed as “Qingke” in China, serves as one nutritious basic food for Tibetan people and a crucial feed source for herds in the QTP [29,30]. The high-altitude ecosystems of the QTP endow Tibetan hulless barley with remarkable resistance to abiotic stresses, characterized by strong cold resistance, a short growing period, high productivity, early maturation, and remarkable adaptability to diverse environments, which makes it an excellent crop model for the analysis of the drought tolerance mechanism [31,32]. In recent years, RNA-seq has been widely employed to unravel the molecular mechanisms underlying drought resistance in hulless barley [33,34]. Comparative transcriptomic investigation across a gradient of water stress treatments in hulless barley leaf tissues and uncovered coordinated regulation by numerous TF families and plant hormone signaling cascades during drought adaptation [35]. As a typical model for extreme environment adaptation, Tibetan hulless barley possesses abundant stress-resistant genetic resources, which establishes its value across multiple research regions [36]. In the early stages of exploring drought resistance, researchers had yet to gain a deep understanding of how Tibetan hulless barley’s lncRNAs regulate gene expression to withstand drought stress in the face of extreme weather conditions. Consequently, elucidating the molecular mechanisms underlying drought response in Tibetan hulless barley holds significant theoretical value for understanding the biological basis of crop adaptation to extreme environments, while also providing novel strategies to address agricultural challenges posed by global climate change.

In this study, we performed a stringent identification of lncRNAs to examine their expression profiles in two cultivars of Tibetan hulless barley subjected to various drought treatments. We characterized the fundamental features of these lncRNAs and identified those that respond to drought stress. Furthermore, we functionally annotated the *cis*- and *trans*-regulatory roles of the target genes of these lncRNAs and predicted their potential functions in drought response. From an applied perspective, this research facilitates the discovery of unique drought-resistant genetic resources in plateau crops, providing valuable targets for molecular design breeding. With increasing global aridity, understanding the adaptation strategies of highly stress-tolerant hulless barley holds significant practical implications for developing climate-smart crops and ensuring food security. Therefore, in-depth investigation of the molecular mechanisms underlying drought response in Tibetan hulless barley not only carries substantial theoretical scientific value but also offers potential solutions for sustainable agricultural development.

## 2. Materials and Methods

### 2.1. The Plant Materials and Growth Conditions

In this study, the Tibetan hulless barley RNA-sequencing raw data were obtained from the previous study published by Liang et al. [35]. Their sequencing of plant materials used the drought-tolerant barley cultivar (Z772) and the drought-sensitive cultivar (Z013). In addition, we planted the drought-sensitive cultivar Kunlun 14 (germplasm repository of the Qinghai Academy of Agricultural and Forestry Sciences) as experimental materials for qRT-PCR validation. To enhance the seed germination rate and growth speed, the seeds were planted in pots filled with a 5:1 mixture of nutrient soil and vermiculite. To ensure optimal growing conditions, we maintained controlled environmental parameters in the growth chamber, with temperatures kept between 23 °C and 25 °C, humidity maintained at 50% to 70%, and a photoperiod of 16 h/8 h light/dark.

### 2.2. Comprehensive Transcriptome Alignment and Assembly

The paired-end RNA-seq raw data for the Tibetan hulless barley with drought response experiment were downloaded from the NCBI Sequence Read Archive (SRA) under the accession number PRJNA391461 [37]. In that study, leaf samples from 10 individuals per accession were pooled from two Tibetan hulless barley isogenic lines, exposed on filter paper for 0, 1, or 5 h (*n* = 6 pools total), flash-frozen in liquid nitrogen, and stored at −80 °C prior to analysis. Then, RNA was extracted from leaf samples and used for library construction and sequencing [37]. Fastp (version 0.23.4; https://opengene.org/fastp/fastp accessed on 19 September 2023; -l 20, -q 5, -u 50) was used to perform quality control (QC) on the initial reads and to clean up the raw data [38]. Furthermore, we selected the hulless barley genome (assembly version: GCA_004114815.1) carried out in 2020, which served as a reference, and utilized the Bowtie2 (version 2.5.1, available at https://bowtie-bio.sourceforge.net/bowtie2/index.shtml, accessed on 28 May 2024) for subsequent analysis [39,40]. This genomic dataset provided an invaluable reference foundation for conducting more precise investigations in our later studies. Additionally, TopHat2 (version 2.0.14; available at https://ccb.jhu.edu/software/tophat/index.shtml, accessed on 28 May 2024; -I 5000) was used for genome index building, a critical step that facilitates rapid and efficient searching of large genomic databases, and for read alignment, a process essential for comparing and analyzing genomic sequences [41]. To achieve transcript assembly and combination, we utilized Cufflinks (version 2.2.1; available at https://cole-trapnell-lab.github.io/cufflinks/install/, accessed 28 May 2024) [42]. Subsequently, the expression levels of the isoforms were quantified using Fragments Per Kilobase of transcript per Million mapped reads (FPKM) to ensure a standardized basis for comparison.

### 2.3. Workflow for Systematic LncRNA Annotation

To determine the molecular characteristics of lncRNAs, we developed a rigorous filtering pipeline (Figure 1a). The transcripts classified as “i” (fully enclosed within a reference intron), “o” (exonic overlaps with a reference transcript), “u” (intergenic transcripts), “j” (potentially novel isoforms), and “x” (exonic overlaps on the opposite strand relative to the reference) were segregated and advanced to the subsequent filtering pipeline. After filtering for transcripts greater than 200 nt, we utilized the BLAST (version 2.13.0) program to identify potential messenger mRNAs, transfer RNAs (tRNAs), and small nucleolar RNAs (snoR-NAs), applying a stringent e-value threshold of less than 10^−10^. With the aid of HMMER [43], the remaining transcripts were used to eliminate those that contained known protein domains outlined in the Pfam database (e-value < 10^−10^) [44]. Moreover, the Coding Potential Calculator (CPC2 version 2.0) and LncRNA Gene Classifier (LGC version 1.0) were combined to assess the noncoding capacity of the transcripts [4,45]. To ascertain whether any of the identified lncRNAs could potentially produce microRNAs (miRNAs), we performed a BLAST search using the candidate lncRNA transcripts against both the miRNA sequences in the hulless barley genome and the mature miRNA sequences listed in the miRbase database. Following this analysis, we further filtered the results, retaining only those isoforms with an FPKM value of 0.5 or higher (FPKM status = OK), thereby ensuring that the final set of lncRNAs was reliably expressed.

### 2.4. Prediction of Trans-/Cis-Regulatory Target Genes of LncRNAs in Hulless Barley

Genes transcribed within a 100 kb window upstream or downstream of lncRNAs were identified as potential *cis*-acting target genes using Bedtools v2.0 software. The study extracted all DEPCGs around lncRNAs in hulless barley using TBtools (version 2.0) [46,47]. The study analyzed all protein-coding genes (PCGs) that were up- or downregulated to compare the expression patterns of *cis*-acting target genes with their corresponding differentially expressed long noncoding RNAs (DElncRNAs). In this study, we conducted a comprehensive analysis of all upregulated differentially expressed protein-coding genes (DEPCGs) and all downregulated DEPCGs. Furthermore, we comprehensively identified all the *cis*-regulated target genes associated with both upregulated and downregulated DElncRNAs. The PCGs and lncRNAs were merged, and the expression level matrix was input into the Weighted Gene Co-expression Network Analysis (WGCNA) package in TBtools [48]. A non-directional co-expression network was constructed within each module, and PCGs that exhibited co-expression relationships with lncRNAs were considered *trans*-regulated target genes. The protein sequence of Tibetan hulless barley was annotated using eggNOG-Mapper (http://eggnog-mapper.embl.de/, accessed on 8 April 2024) as a reference to understand the functions of the *cis*-/*trans*-regulated target genes [49]. TBtools was used to conduct Gene Ontology (GO) enrichment analyses on the *cis*- and *trans*-regulated target genes, employing a significance threshold of FDR < 0.05.

### 2.5. Differential Expression Analysis

Cuffdiff was utilized to calculate the FPKM values for lncRNAs. Subsequently, the FPKM method was applied to determine the expression levels of each transcript. DElncRNAs and DEPCGs were conducted in the drought-tolerant cultivar (Z772) and the drought-sensitive cultivar (Z013) throughout their developmental timeline, using FPKM values. In this study, a log2 fold change value of 1 or greater, combined with a *p*-value of 0.05 or less, was used to identify differentially expressed genes [50].

### 2.6. Drought Stress Treatments

Twenty-five-day-old Kunlun 14 seedlings were selected for drought stress treatment. The intact fifth leaves (exhibiting vibrant green coloration without visible lesions) were excised using RNAse-free scissors and placed on sterile filter paper. Leaf samples were subjected to controlled dehydration at 23–25 °C with 51% relative humidity for 0, 1, or 5 h durations. Equal amounts of leaves from three biologically independent individuals (biological replicates, *n* = 3) were collected, pooled by time point, and subjected to filter paper treatment for 0 h, 1 h, and 5 h, respectively. These nine samples were quickly ground using a mortar and pestle with liquid nitrogen and then stored in a −80 °C freezer.

### 2.7. Total RNA Extraction, cDNA Construction, and Quantitative Real-Time Polymerase Chain Reaction (qRT-PCR) Validation

To validate the results of RNA-Seq, two pairs of DElncRNA-PCG were chosen as targets for qRT-PCR analysis. Firstly, total RNA was extracted from five-leaf stage tissues of Kunlun 14 using Trizol reagent (Thermo Fisher Scientific, Waltham, MA, USA. Cat. 15596026CN), following the manufacturer’s instructions. The quality of the extracted RNA was assessed using a Nanodrop 2000 spectro-photometer (Thermo Fisher Scientific, Cat.). cDNA synthesis was performed using the EasyScript ^®^ First-Strand cDNA Synthesis SuperMix (TransGen Biotech Co., Ltd., Beijing, China) as per the manufacturer’s guidelines. The first-strand cDNA was synthesized using 1000 ng of RNA samples, diluted tenfold, and 1 μL was used in a 20 μL PCR reaction system. The PCR amplification consisted of a preincubation at 95 °C for 5 min and 40 cycles, each comprising 15 s at 95 °C, 15 s at 60 °C, and 15 s at 72 °C. The reactions utilized the QuantStudio real-time PCR system (Bio-Rad, Boston, MA, USA) and iQ SYBR Green Supermix (Bio-Rad Laboratories, Shanghai, China). To standardize the cDNA templates, the housekeeping gene Elongation Factor 1-alpha (*EF1α)* was co-amplified. All primers were synthesized by RuiBiotech (Beijing, China) and are presented in Appendix A.

## 3. Result

### 3.1. Genome-Wide Scale Transcriptional Signatures of LncRNAs in Two Tibetan Hulless Barley Cultivars Under Drought Conditions

With the assistance of a chromosome-scale genome assembly for Tibetan hulless barley [51], we further explored the published transcriptomic data of the drought-tolerant cultivar (Z772) and the drought-sensitive cultivar (Z103) under various drought-stress treatment durations [35] to reveal drought-induced lncRNA expression profiles. Following a rigorous filtering pipeline including length selection, coding potential assessment, and expression level evaluation, we identified 2877 lncRNAs from a total of 133,197 transcripts in the six Tibetan hulless barley samples (Figure 1a and Appendix A). The lengths of the lncRNAs ranged from 201 to 5837 nucleotides (nt), with an average length of 596 nt (Figure 1b and Appendix A). Analysis of exon distribution patterns of lncRNAs has revealed a striking predominance of single-exon transcripts, with more than half (1664 out of 2877, 57.8%) of the identified lncRNAs being single-exonic, in contrast to 24.0% (691) that contained two exons, with only 18.1% (522) exhibiting more than three exons (Figure 1c and Appendix A). Additionally, the 2877 lncRNAs were classified into four groups based on their class_code, which includes 2812 intergenic lncRNAs (lincRNAs, u), 30 antisense lncRNAs (x), and 30 sense lncRNAs (further divided into 21 o and 9 j), depending on the relative positions of the transcripts to the PCGs in the barley genome. These results showed that Tibetan hulless barley contains multiple categories of lncRNAs, with lincRNAs being the most dominant form (Figure 1d and Appendix A). Intriguingly, while intronic lncRNAs have been reported in other species [52], none were detected in Tibetan hulless barley, suggesting potential species-specific lncRNA features that may relate to high-altitude adaptation.

### 3.2. Characterization of DElncRNAs in Two Hulless Barley Cultivars Under Different Drought Treatment Times

To further delineate the potential functions of lncRNAs in two hulless barley cultivars under drought stress, we utilized FPKM values of lncRNAs to investigate their expression patterns. Then, the DElncRNAs were screened, which may be involved in the drought stress response. In order to distinguish the different samples of barley, the samples of the drought-tolerant cultivar (Z772) after 0, 1, and 5 h of drought treatment were designated as T0, T1, and T5, and samples of the drought-sensitive cultivar (Z013) subjected to the same treatment durations are labeled S0, S1, and S5, respectively. Upset and Venn diagrams were employed to uncover the co-expression patterns of lncRNAs between the two cultivars. Then, we found that 2179 lncRNAs were co-expressed in two hulless barley cultivars, whereas 331 and 367 lncRNAs were specifically expressed in each, respectively (Figure 2a and Appendix A). The specifically expressed lncRNAs appear to be directly related to the drought tolerance of the cultivars. The co-expressed lncRNAs may play conserved regulatory roles in both cultivars. Based on hierarchical clustering of the lncRNA expression matrix and their expression patterns, T0, T1, and T5 were clustered into one group, whereas S0, S1, and S5 formed another group (Figure 2b and Appendix A). A stringent criterion we established: log2 (fold change) values ≥ 1, *p* value ≤ 0.01 to elucidate DElncRNAs, and a total of 1004 DElncRNAs were found in this study (Appendix A). A more detailed analysis revealed that the T0/T1 and S1/S5 groups exhibited closer clustering than the T5 and S0 groups, indicating that mild drought and control conditions showed greater expression pattern concordance in drought-tolerant cultivars. Conversely, in drought-sensitive cultivars, lncRNAs demonstrated similar expression patterns under two different levels of drought stress.

### 3.3. Exploring the Regulatory Mechanisms (Trans- and Cis-) of LncRNAs Based on the Distances Between PCGs and LncRNAs

It is clear that plant lncRNAs regulate the growth and developmental processes through both *cis*-acting (local) and *trans*-acting (distal) mechanisms [53]. We first systematically investigated the *trans*-regulatory interactions among all PCGs and the identified lncRNAs in two hulless barley cultivars (Z772 and Z013) under different treatment times. We subjected the screened 2877 lncRNAs and 38,453 PCGs to WGCNA to infer the *trans*-regulatory roles of lncRNAs by utilizing the biological functions of their co-expressed PCGs. After parameter optimization, 923 lncRNAs and 684 PCGs were enriched for association with drought stress for module identification. Hierarchical clustering revealed 11 highly correlated modules (Figure 3a and Appendix A). It is noteworthy that the turquoise, blue, brown, purple, and green modules displayed a significantly higher number of co-clustered lncRNAs and PCGs when compared to the remaining modules. Despite the considerable presence of co-expressed PCGs, lncRNAs were comparatively scarce within each module. Furthermore, module–trait relationships were also discernible in our study (Figure 3b). Following this, we chose two modules of particular interest, “Brown” and “Yellow,” for more in-depth exploration.

The “Yellow” module refers to the time response to drought stress. In the “Yellow” module, following 5 h of drought treatment, the gene exhibited upregulated expression in both hulless barley cultivars, indicating that prolonged drought stress initiates its responsive regulation (Figure 3c). The results of the GO analysis for PCGs in this module revealed a significant enrichment in biological processes related to “response to oxygen-containing compounds”, “response to stress”, and “response to abiotic stimulus”, further corroborating the observations made in the “Yellow” module (Figure 3d). This suggests that prolonged drought stress triggers an emergency transcriptional reprogramming in Tibetan hulless barley, with potential cascading effects on primary metabolic pathways essential for stress survival.

The “Brown” module is called the drought stress cultivar-response module. In the “Brown” module, we observed a significant upregulation in the expression of both lncRNAs and PCGs at 1 h of drought treatment in the drought-sensitive cultivar, higher than all other treatment groups and time points (Figure 3e). GO enrichment analysis of this module PCGs revealed their primary functional associations with “water deprivation response”, “abiotic stress response”, “regulation of metabolic processes”, and “positive regulation of macromolecule biosynthetic and metabolic processes” (Figure 3f). The co-expression network analysis identified the lncRNAs and PCGs that are involved in the response of Tibetan hulless barley to drought stress, shedding light on their potential functions. The results suggest that drought-sensitive cultivars exhibit an earlier activation of stress-responsive mechanisms compared to drought-tolerant cultivars under water deficit conditions, providing valuable insights into the molecular basis of drought tolerance in barley.

Beyond their well-characterized *trans*-regulatory roles, lncRNAs participate in *cis*-regulation to modulate transcriptome dynamics. We obtained the potential target PCGs located within a 100 kilobase (kb) upstream and downstream of the 91 screened DElncRNAs (maintained differential expression patterns across all drought stages). GO enrichment analysis demonstrated significant enrichment in key biological processes including “organic hydroxy compound metabolic process”, “regulation of RNA biosynthesis”, “response to stress”, and “gene expression” (Figure 3g). Notably, both *cis*- and *trans*-regulatory mechanisms showed overlapping enrichment in stress-related pathways, suggesting that hulless barley lncRNAs may orchestrate drought stress responses through dual regulatory modes.

### 3.4. Expressional Dynamics of DElncRNAs in Two Cultivars Across Drought Stress Treatments

To investigate the expression profiles of lncRNAs in Tibetan hulless barley, 1004 DElncRNAs were identified between two cultivars during three different drought treatments. We found the number of upregulated DElncRNAs significantly exceeding that of downregulated DElncRNAs across all treated stages (Figure 4a and Appendix A). When compared in two cultivars, the drought-tolerant cultivar shows 50 co-upregulated DElncRNAs (Figure 4b) and 10 co-downregulated DElncRNAs with the drought-sensitive cultivar under 1 h drought treatment (Figure 4c). Furthermore, following a 5 h drought treatment, 26 co-upregulated (Figure 4d) and 11 co-downregulated DElncRNAs were observed (Figure 4e; all the up- and downregulated lncRNAs mentioned above are listed in Appendix A). Subsequently, we focused on analyzing the dynamics of DElncRNAs in the two cultivars separately. In the drought-tolerant cultivar, 64 DElncRNAs were co-upregulated (Figure 4f), and 10 were co-downregulated after 1 and 5 h of drought treatments (Figure 4g). In contrast, the drought-sensitive cultivar exhibited 78 co-upregulated DElncRNAs (Figure 4h), and 10 were co-downregulated (Figure 4i; the data are listed in Appendix A). These findings demonstrate that numerous lncRNAs display stage-specific responsiveness to drought stress, exhibiting markedly divergent expression profiles between mild and severe dehydration conditions.

### 3.5. The Regulatory Diversity of DElncRNAs and Nearby Potential Target DEPCGs in Two Cultivars Across Drought Stress Treatments

We identified 85 DElncRNAs with potential target DEPCGs in two cultivars of hulless barley under different drought conditions (Figure 5a and Appendix A). We further discovered several DElncRNAs with similar expression patterns between the two cultivars. For instance, TCONS_00004693, TCONS_00093746, TCONS_00128626 (Figure 5b), TCONS_00035694, and TCONS_00051036 (Figure 5c) demonstrated downregulated expression under drought stress in both drought-tolerant and drought-sensitive cultivars. Meanwhile, TCONS_00070789, TCONS_00082545 (Figure 5c), and TCONS_00023895 (Figure 5d) showed no significant response to mild drought stress but were markedly upregulated under severe drought stress in both cultivar types. Additionally, TCONS_00024004 (Figure 5b), TCONS_00039473, TCONS_00079580, TCONS_000107383 (Figure 5d), and TCONS_00020976 (Figure 5e) were upregulated under mild drought stress but downregulated under severe drought stress in both cultivars. We also examined the expression patterns of DEPCGs and observed that the majority of DEPCGs in the drought-tolerant cultivar exhibited no significant changes in expression levels after 5 h of drought treatment, whereas the expression levels were significantly altered in the drought-sensitive cultivar (Figure 5f and Appendix A). This pattern is similar to that observed for DElncRNAs. The results suggest that these lncRNAs and their target DEPCGs exhibit distinct expression patterns across the six samples, highlighting their functional diversity in regulatory processes.

### 3.6. In-Depth Functional Profiling of Putative LncRNA Target Genes

We further analyzed the expression patterns of DElncRNA-PCG pairs and identified 12 pairs with similar expression profiles, including 5 pairs of drought stress downregulated DElncRNA-PCGs, respectively: the TCONS_00118699-D1007_56089 pair (with no annotations provided); the TCONS_00099675-D1007_46973 pair functions as an anion transporter 2; TCONS_00051036-D1007_24041 encodes cellulose synthase 8 (CesA8); TCONS_00004693-D1007_02105 encodes a ZF-HD transcription factor; and the TCONS_00020976-D1007_09804 pair functions as putative disease resistance protein Resistance Gene Analog 1 (Figure 6a). Four pairs of DElncRNA-PCG pairs were highly upregulated following 5 h of drought treatment: TCONS_00023895-D1007_11257 (with no annotation provided); TCONS_00060403-D1007_28477, encoding a stress-responsive MYB3 protein; TCONS_00122975-D1007_58167, encoding a gliadin protein; and TCONS_00026388-D1007_12461, encoding a protein known as MAD1-like. Additionally, three pairs of DElncRNA-PCG pairs were upregulated after 1 h of drought treatment but downregulated after 5 h, respectively: TCONS_00071826-D1007_33860 encoding Glutathione S-Transferase Tau 6; TCONS_00108852-D1007_51314 (none annotated); TCONS_00055204-D1007_25929 (Ethylene response factor 109, ERF 109-like) (Figure 6c; Details of the 12 selected pairs are presented in Appendix A). Our study demonstrates that specific lncRNAs modulate the response of Tibetan hulless barley to drought stress through the regulation of transcription factors (TFs), thereby playing a crucial role in the adaptive mechanisms of this plant species to environmental challenges.

Moreover, we utilized the drought-sensitive cultivar, Kunlun 14, to verify the reliability of the RNA-Seq data. Two DElncRNA-PCG pairs (TCON_00020976-D1007_09804 and TCON_00055204-D1007_25929) were selected for qRT-PCR analysis (Figure 6d). TCONS_00055204 and D1007_25929 were dramatically upregulated following 1 h of drought treatment (Figure 6e). Conversely, TCONS_0020976 and D1007_09804 were significantly upregulated after 1 h but downregulated after 5 h of drought treatment (Figure 6e). The expression patterns of these two pairs of DElncRNA-PCGs were in remarkable accordance with the trends evident in the transcriptomic data. The qRT-PCR results served to confirm the expression patterns of these genes, exhibiting a robust correlation with the RNA-Seq data, thereby upholding the reliability of the data and ensuring the authenticity of the differentially expressed genes.

## 4. Discussion

Drought, a critical abiotic stressor, severely compromises agricultural productivity by disrupting essential physiological and biochemical processes in plants [54]. Among staple grains, species such as Tibetan hulless barley are particularly vulnerable to drought stress [55]. However, its simple genome and exceptional stress resilience render Tibetan hulless barley a more reliable system for studying genetic and molecular mechanisms of drought tolerance compared to other cereal crops, establishing it as an ideal model for dissecting drought adaptation in cereals [56]. Emerging evidence highlights the pivotal roles of lncRNAs in mediating drought-responsive biological processes across plant species [10,57]. For example, in *Arabidopsis*, the lncRNA DANA1 enhances drought tolerance by interacting with DIP1 and PWWP3 [58]. Despite these advances, the functional landscape of lncRNAs in Tibetan hulless barley’s drought response remains largely unexplored, primarily due to limited functional validation. Our study addresses this gap by systematically investigating the molecular mechanisms through which lncRNAs confer drought stress tolerance in Tibetan hulless barley.

The advent of high-throughput technologies has significantly advanced the field of plant lncRNA research [34,35,36], leading to the identification of numerous lncRNAs across plant genomes, including 2937 in rice [59], 1624 in *Orinus* [60], and others [27,61]. In this study, we applied a stringent lncRNA screening pipeline to identify 2877 lncRNAs (Figure 1a) in the Tibetan hulless barley genome (Assembly: GCA_004114815.1) [39]. Structural analysis revealed that these lncRNAs are predominantly short (Figure 1b), single-exon transcripts of the intergenic type (Figure 1c). Notably, hulless barley lncRNAs exhibit fewer exons compared to other Poaceae species [62,63]. We hypothesize that this structural simplification may reflect evolutionary adaptations to high-altitude environmental pressures. Future studies will systematically explore this phenomenon to elucidate its underlying evolutionary mechanisms and functional consequences.

Analysis of GO enrichment and co-expression networks (Figure 3f,g) revealed that drought-responsive lncRNAs coordinate metabolic adaptation in hulless barley through three key processes: osmolyte metabolism (GO: “organic hydroxy compound metabolic process”), transcriptional regulation (GO: “RNA biosynthesis”), and stress response (GO: “response to stress”). Tolerant cultivars exhibited optimized metabolic reprogramming that maintained essential functions, while sensitive lines showed premature, resource-intensive activation (Figure 3f). These findings demonstrate lncRNA-mediated metabolic plasticity as central to drought adaptation, with potential applications in metabolic engineering for crop improvement.

As demonstrated in prior studies, lncRNAs serve crucial roles of transcriptional activation and repression across both plant and animal systems [64,65]. In this study, we focused specifically on analyzing the expression patterns of DElncRNAs in Tibetan hulless barley. Under moderate drought stress, both hulless barley cultivars exhibited comparable numbers of upregulated genes. However, under extreme drought conditions, pronounced differences in DElncRNAs expression emerged: the drought-tolerant cultivar showed significantly more upregulated DElncRNAs than the drought-sensitive cultivar. Remarkably, under moderate drought stress, both hulless barley varieties showed comparable numbers of upregulated genes. However, under severe drought conditions, striking differences in DElncRNAs emerged: the drought-tolerant cultivars exhibited significantly more upregulated DElncRNAs than the drought-sensitive cultivar (Figure 4). This observation contrasts markedly with Liang et al.’s findings that demonstrated the progressive upregulation of PCGs in drought-sensitive cultivars under intensifying stress conditions [35]. Furthermore, the outcomes of the GO enrichment analysis exhibit marked differences as well. In Liang’s research, DEPCGs were discovered to play pivotal roles in modulating protein synthesis and energy metabolism. In contrast, our analysis of neighboring genes indicates that lncRNAs mainly target pathways responsive to water and stress. Nevertheless, this study indicates that lncRNAs in the drought-sensitive cultivar employ distinct regulatory mechanisms compared to PCGs during the process of adapting to water stress.

Furthermore, our analysis revealed that lncRNA-targeted PCGs exhibit functionally significant associations with key transcription factors (TFs), including ERF, MYB, and ZF-HD families (Figure 6), which play pivotal roles in mediating the epigenetic regulation of drought-responsive pathways [66,67,68,69]. In transgenic tomatoes and apples, a factor associated with the AP2/ERF family, designated as MhERF113-like, is believed to function as a positive regulator in enhancing drought tolerance [70]. Notably, *MYB41-BRAHMA* (a canonical MYB family member) [71] and *PtrVCS2* (a ZF-HD family protein) [72] have been shown to enhance drought tolerance in *Arabidopsis* and *Populus trichocarpa* through the regulation of stomatal movement and xylem vessel morphology, respectively. These findings underscore the importance of investigating key TFs in Tibetan hulless barley to elucidate its drought adaptation mechanisms. Intriguingly, we observed distinct expression patterns for two identified TFs—MYB (D1007_28477) and ERF (D1007_25929)—between the two cultivars: both exhibited time-dependent upregulation in the drought-sensitive cultivar under drought stress, while remaining stable in the tolerant cultivar. This differential regulation suggests these TFs may function specifically in the drought-inducible response of sensitive plants. Conversely, the drought-tolerant cultivar appears to maintain a constitutive genetic program that sustains osmotic balance and oxygen species (ROS) scavenging capacity even under normal conditions, potentially obviating the need for stress-induced TFs activation [34,73]. We posit that these transcription factors engage with the cis-acting elements of downstream target genes, subsequently modulating their expression. This regulatory mechanism holds significant sway over plant growth, development, and the plant’s response to stress. These disparate strategies underscore the imperative for additional comparative studies aimed at comprehensively unraveling the molecular underpinnings of drought resistance divergence prevalent among these Tibetan hulless barley cultivars.

The temporal expression profiling identified distinct drought-responsive lncRNA dynamics, with TCONS_00055204 and its putative ERF-family TF target D1007_25929 showing rapid 1 h upregulation (Figure 6e), implicating their involvement in early stress signaling, while TCONS_00020976 and D1007_09804 displayed transient 1 h peaks followed by decline (Figure 6e), suggesting roles in osmotic response modulation. While these short-term (1–5 h) data effectively characterize initial molecular responses, they incompletely represent the full spectrum of drought adaptation mechanisms that typically emerge over longer timescales. The physiological significance of these transient expression patterns remains to be established through extended duration studies (72 h to weeks) that would better correlate molecular events with adaptive phenotypes under sustained drought conditions.

Although this study has not yet experimentally validated the specific functions of lncRNAs, systematic bioinformatics analyses and expression pattern studies have provided insights into their potential mechanisms in drought response in Tibetan hulless barley. The observed specific co-expression patterns between the identified lncRNAs and known drought-responsive transcription factors (such as members of the ERF and MYB families) suggest they may influence downstream target gene expression by regulating the formation of transcriptional complexes or chromatin states. These lncRNAs generally exhibit structural simplicity and dynamic expression changes, characteristics that may correspond to their functional requirements for rapid responses to environmental stress. Notably, the differential expression patterns of lncRNAs between drought-tolerant and drought-sensitive varieties, along with their time-dependent induction features following drought treatment, further support the hypothesis that lncRNAs may serve as key nodes in regulatory networks involved in barley’s drought adaptation. Although these functional predictions require further experimental validation, this study provides important clues for elucidating the molecular mechanisms of lncRNAs in crop abiotic stress responses.

Our study represents the first comprehensive exploration of lncRNAs in the context of Tibetan hulless barley’s response to drought stress. Through our research, we identified a significant array of drought-responsive lncRNAs and predicted their likely target genes, many of which are engaged in diverse biological processes. Our findings suggest that the exceptional drought tolerance exhibited by Tibetan barley may stem from the adaptive co-evolution of MYB and ERF TFs with other transcriptional regulators. While this study establishes important molecular insights into lncRNA-mediated drought responses in Tibetan hulless barley, several limitations should be acknowledged. The hypothetical nature of lncRNA functions without genetic validation, potential discrepancies between controlled and field drought conditions, and unresolved evolutionary questions regarding lncRNA structural simplification represent key areas for future investigation. These findings nevertheless provide a valuable foundation for understanding ncRNA contributions to drought resilience, identifying promising targets for crop improvement strategies. Further research integrating functional genomics and field-based approaches will be essential to fully elucidate the complex drought tolerance mechanisms in this important cereal crop.

## 5. Conclusions

This study examines the potential involvement of lncRNAs in drought-tolerance mechanisms in hulless barley. RNA-seq analysis comparing a drought-tolerant (Z772) and a drought-sensitive (Z013) hulless barley cultivar identified 2877 putative lncRNAs. Of the 2877 identified lncRNAs, 1663 (57.8%) were single-exon transcripts, and no intron-containing lncRNAs were observed. We detected 2179 shared lncRNAs between cultivars, along with 331 drought-tolerant-specific and 367 drought-sensitive-specific lncRNAs. Furthermore, we identified a total of 22 lncRNAs that were differentially expressed across all treatment conditions. For practical applications, the identified 22 drought-responsive lncRNAs and their regulatory targets (particularly TFs) could be exploited through molecular marker-assisted selection or CRISPR-Cas9 genome editing to breed superior drought-resistant hulless barley cultivars. WGCNA identified 11 modules enriched in drought-responsive pathways. Additionally, the target genes of *trans*-/*cis*-regulated lncRNAs predominantly exhibited functions related to stress responses. Twelve pairs of DElncRNA-PCG were discovered to exhibit identical expression patterns, and a number of PCGs were identified to function as TFs, suggesting their possible roles as cis-regulatory elements in pathways responsive to drought stress. qRT-PCR validated key RNA-seq results. We constructed a Tibetan hulless barley lncRNA database—the first drought-responsive catalog for this species—providing insights into lncRNA regulation. Future work combining multi-omics and gene editing may advance practical applications.

## Figures and Tables

**Figure 1 biology-14-00737-f001:**
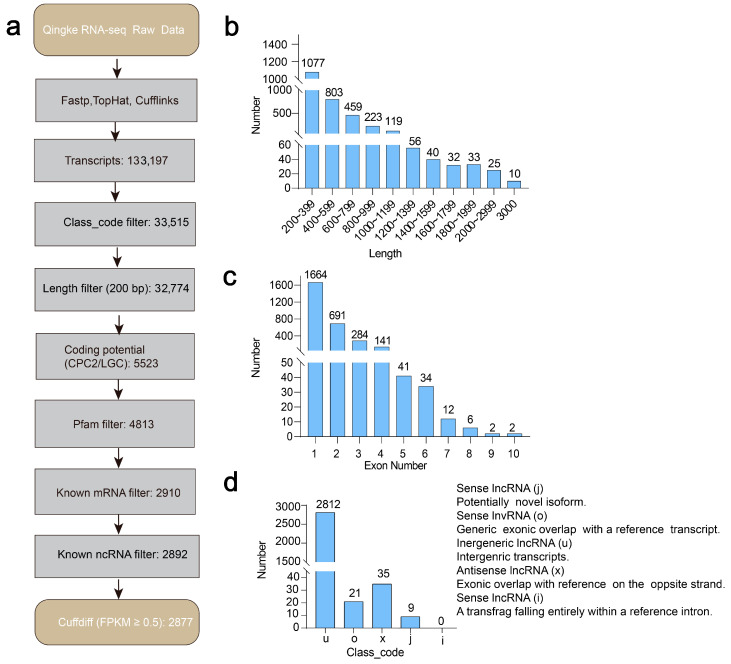
Exhaustive spectrum and identification of long noncoding RNAs (lncRNAs) cross the RAN-seq of hulless barley. (**a**) Screening tube for lncRNAs. (**b**) Molecular length characteristics of lncRNAs. (**c**) Distribution pattern of exon numbers among lncRNAs. (**d**) Allocation of lncRNAs class_code (i, o, u, j, x). The lncRNAs were classified according to the location of the lncRNAs compared with the annotated genes in the genome.

**Figure 2 biology-14-00737-f002:**
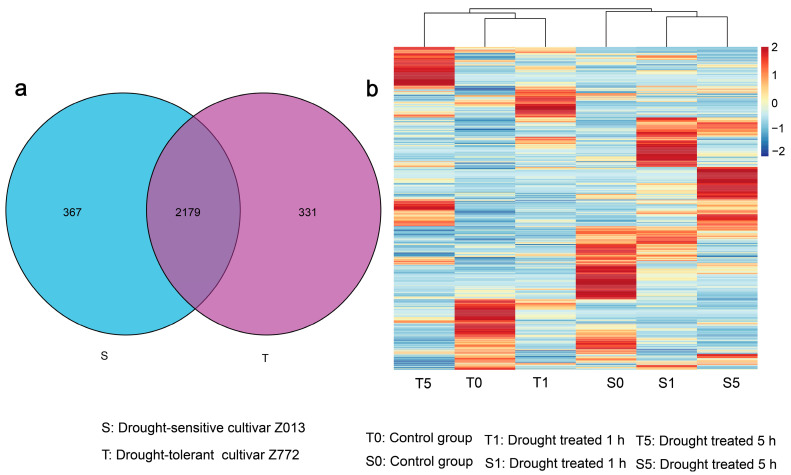
The expression distribution and patterns of lncRNAs in two hulless barley cultivars (Z772 and Z013) under different treatment times. (**a**) Unique and co-expressed lncRNAs in the two cultivars. (**b**) The heat map of lncRNAs under different drought treatment durations.

**Figure 3 biology-14-00737-f003:**
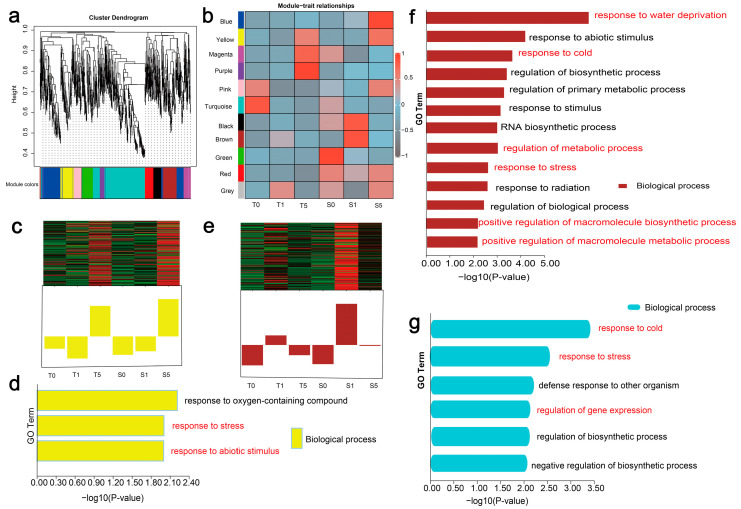
Comprehensive analysis of the *trans*-/*cis*-acting functions of lncRNAs in hulless barley under drought stress. (**a**) Hierarchical cluster tree and color bands showing the 11 modules by WGCNA (*n* = 1607). (**b**) The statistical analysis of module–trait correlations. The rows and columns indicate the modules and traits, respectively. (**c**) Eigengene expression profile for the “Yellow” module of hulless barley in response to drought. (**d**) Terms of GO enrichment analysis among PCGs and their *trans*-regulated lncRNAs of the “Yellow” module. (**e**) Eigengene expression profile for the “Brown” module of hulless barley in response to drought. (**f**) Terms of GO enrichment analysis among PCGs and their *trans*-regulated lncRNAs of the “Brown” module. (**g**) Terms of GO enrichment analysis among *cis*-regulated target genes of DElncRNA. In the GO enrichment analysis figure:Yellow represents the yellow module of lncRNAs and PCGs clustered by WGCNA. Brown represents the brown module of lncRNAs and PCGs clustered by WGCNA. Turquoise indicates the GO enrichment results of cis-regulated PCGs.

**Figure 4 biology-14-00737-f004:**
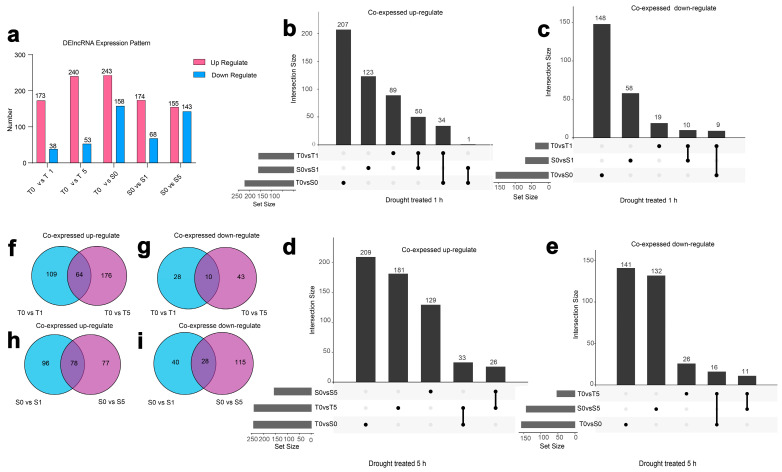
Expression profiles of DElncRNAs across Tibetan hulless barley cultivars under differential drought treatments. (**a**) Expression patterns of DElncRNAs under drought stress in two hulless barley cultivars. The Upset graph indicated the count of unique and shared (**b**) upregulated and (**c**) downregulated DElncRNAs between these three groups after 1 h drought treatment, and (**d**) Upregulated DElncRNAs after 5 h drought treatment, (**e**) downregulated after 5 h drought treatment. The Venn diagram displayed the amount of (**f**) upregulated and (**g**) downregulated DElncRNAs in the drought-tolerant cultivar after 1 h and 5 h drought treatments, and (**h**) upregulated and (**i**) downregulated after 1 h and 5 h drought treatments in drought-sensitive cultivar.

**Figure 5 biology-14-00737-f005:**
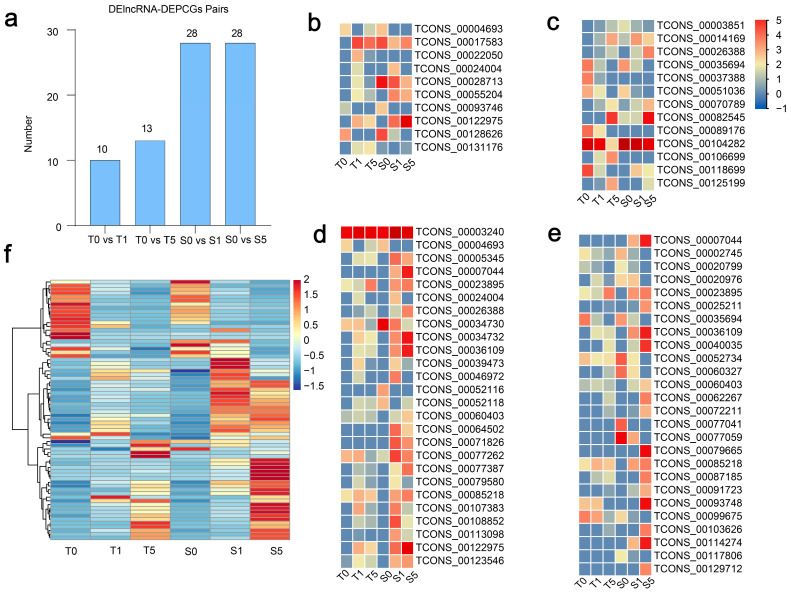
Expression patterns of DElncRNA-PCG pairs. (**a**) Number of DElncRNA-PCG pairs in different treatment groups. (**b**) Expression levels of DElncRNAs in the drought-tolerant cultivars under 0 h and 1 h drought treatment conditions, (**c**) under 0 h and 5 h drought treatment conditions, (**d**) under 0 h and 1 h drought treatment conditions, and (**e**) under 0 h and 5 h drought treatment conditions. (**f**) The heatmap of DEPCGs of DElncRNA.

**Figure 6 biology-14-00737-f006:**
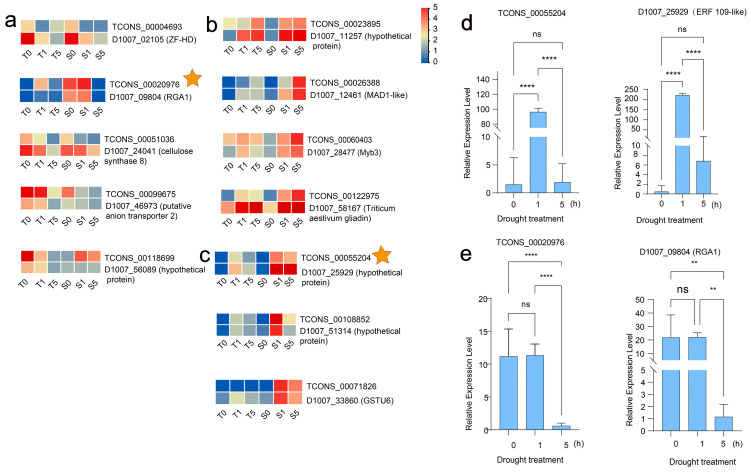
Expression profile of 12 co-expressed lncRNA-PCGs pairs under different drought treatment conditions. (**a**) Five pairs of drought stress downregulate lncRNA-PCG pairs of drought-induced downregulated lncRNAs, which form lncRNA-PCGs. (**b**) Four highly upregulated pairs after 5 h of drought treatment, which construct lncRNA-PCGs. (**c**) Temporal expression patterns of 3 lncRNA-PCG pairs (upregulated at 1 h but downregulated at 5 h). (**d**) TCONS_00055204-D1007_25929 relative expression level under varying drought durations. (**e**) TCONS_00020976-D1007_09804 relative expression level under varying drought treatments. The expression levels in each sample were calculated in log_2_ (FPKM) units (** *p* < 0.01, **** *p* < 0.0001, ns (not significant, *p* > 0.05); one-way ANOVA with nonparametric or mixed-effects tests; *n* = 3). The genes selected for qRT-PCR validation (TCONS_00020976 from (**a**) and TCONS_00055204 from (**c**)) are marked with star symbols in the figures. The “drought treatment” is defined as ‘duration of drought treatment in detached leaves’.

## Data Availability

The original contributions presented in the study are included in the article/Appendix A, and further inquiries can be directed to the corresponding author/s. The reads are deposited in the Sequence Read Archive (SRA) under SRP111592.

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
