# Peer review of "Transcriptomic Analysis Reveals the Role of Long Non-Coding RNAs in Response to Drought Stress in Tibetan Hulless Barley"

_biology, 2025, doi:10.3390/biology14070737_

Round 1
Reviewer 1 Report
Comments and Suggestions for Authors
Well done.
The manuscript is well-structured, providing detailed results, clear methodologies, and biologically relevant findings; however, it is lengthy, with repetitive information, and lacks a practical application focus. I have some comments and suggestions to improve the manuscript. Kindly find the attached file.

The English could be improved to convey the research more clearly.
Reviewer 2 Report
Comments and Suggestions for Authors
This manuscript explores the role of long non-coding RNAs (lncRNAs) in response to drought stress in Tibetan hulless barley. While the study presents valuable findings, several areas require enhancement in clarity, depth, and presentation. Below are some key comments aimed at improving the manuscript's structure, methodology, and interpretation of results.
- The title should clearly reflect the study’s methodology or key variables to attract the right audience.
- The abstract should be revised to succinctly summarize the methods, results, and conclusions.
- Expand the introduction to include a more thorough literature review, discussing current gaps in the field.
- Clearly state the research question or hypothesis at the end of the introduction.
- Provide a more detailed description of the methodologies, including data collection and analysis techniques.
- Justify the statistical methods used in the study, ensuring they align with the data type and research questions.
- Discuss how the sample size was determined, including any power analysis conducted.
- Define all variables and constructs measured in the study to avoid ambiguity.
- Enhance data presentation with well-labeled tables and figures that contribute meaningfully to the results discussion.
- Interpret results in the context of existing literature to show how the findings add to the current body of knowledge.
- Acknowledge the study's limitations, including biases or methodological constraints.
- Strengthen the conclusion by discussing practical implications and possible future applications.
- Update references with more recent and relevant studies, including: DOI: 10.3389/fpls.2025.1538664, DOI: 10.3390/su15086512, DOI: 10.1201/9781003180579-2.
- Revise the manuscript for grammatical accuracy and clarity, possibly with professional proofreading services.
- Include ethical approval information, along with any consent procedures if applicable.
- Add a statement on potential conflicts of interest to ensure transparency.
- Acknowledge all funding sources and grant numbers in the manuscript.
- Include any supplementary materials (e.g., questionnaires, datasets) in an accessible format.
- Ensure the manuscript adheres to the journal’s formatting guidelines for consistency.
- Clarify the contributions of each author to the research, providing transparency on authorship.
English needs a minor check
Reviewer 3 Report
Comments and Suggestions for Authors
The authors try to identify the long non-coding RNAs in response to drought stress in Tibetan. It provides a big database. However, the study is lack of analysis in depth.
Major points:
- I did not find the original sequencing data. Please upload all the sequencing data in SRA database or other public database.
- The standard used for the differential expression analysis is too low. The padj value should be calculated and used for the analysis.
- The whole study is lack of experiment repeats. I did not find any description of the repeats of this paper.
- It is good to use both drought-resistance and drought-sensitive plants for the study. However, no depth analysis was performed between the different genes with the different patten of these two kinds of plants. This is quite important to find the potential non-coding RNAs related with the drought-stress.
Round 2
Reviewer 3 Report
Comments and Suggestions for Authors
Could you please cite recent papers to use the p value<0.05, log2 FC>=1 for the qualification?
Also, other comments were suggested to improve the quality of this paper. The paper now is still not good enough for publication.
